# A Kerr polarization controller

N. Moroney[1,2], L. Del Bino [1], S. Zhang[1], M. T. M. Woodley [1,2,3], L. Hill [1,4], T. Wildi[5], V. J. Wittwer [6], T. Südmeyer[6], G.-L. Oppo [4], M. R. Vanner[2], V. Brasch[7], T. Herr [5,8] & P. Del'Haye [1,9✉]

Kerr-effect-induced changes of the polarization state of light are well known in pulsed laser systems. An example is nonlinear polarization rotation, which is critical to the operation of many types of mode-locked lasers. Here, we demonstrate that the Kerr effect in a high-finesse Fabry-Pérot resonator can be utilized to control the polarization of a continuous wave laser. It is shown that a linearly-polarized input field is converted into a left- or right-circularly-polarized field, controlled via the optical power. The observations are explained by Kerr-nonlinearity induced symmetry breaking, which splits the resonance frequencies of degenerate modes with opposite polarization handedness in an otherwise symmetric resonator. The all-optical polarization control is demonstrated at threshold powers down to 7 mW. The physical principle of such Kerr effect-based polarization controllers is generic to high-Q Kerr-nonlinear resonators and could also be implemented in photonic integrated circuits. Beyond polarization control, the spontaneous symmetry breaking of polarization states could be used for polarization filters or highly sensitive polarization sensors when operating close to the symmetry-breaking point.

---

[1] Max Planck Institute for the Science of Light, 91058 Erlangen, Germany. [2] QOLS, Blackett Laboratory, Imperial College London, SW7 2AZ London, UK. [3] SUPA and Department of Physics, Heriot-Watt, Edinburgh EH14 4AS, UK. [4] SUPA and Department of Physics, University of Strathclyde, Glasgow G4 0NG, Scotland. [5] Center for Free-Electron Laser Science CFEL, Deutsches Elektronen-Synchrotron DESY, Hamburg, Germany. [6] Laboratoire Temps-Fréquence, Université de Neuchâtel, CH-2000 Neuchâtel, Switzerland. [7] Swiss Center for Electronics and Microtechnology (CSEM), Time and Frequency, Neuchâtel, Switzerland. [8] Physics Department, Universität Hamburg, 22761 Hamburg, Germany. [9] Department of Physics, Friedrich Alexander University Erlangen-Nuremberg, 91058 Erlangen, Germany. ✉email: pascal.delhaye@mpl.mpg.de

Spontaneous symmetry breaking is an important concept in fundamental physics, describing the origins of bosonic mass via the Higgs mechanism[1], superconductivity[2], and the phases of matter[3]. Spontaneous symmetry breaking is characterized by a system whose Lagrangian and initial state are symmetric (invariant under some transformation), but whose lowest-energy states to which the system evolves do not share such a symmetry.

Nonlinear optical interactions and in particular the Kerr effect can also exhibit spontaneous symmetry breaking. An example is time-reversal symmetry breaking in a pulse-pumped ring cavity[4,5]. In addition, the Kerr interaction plays an important role in the interaction of soliton frequency combs in microresonators[6–10]. In the continuous wave regime, spontaneous symmetry breaking has been experimentally observed[11–13] between counter-propagating light in microresonators with high optical quality factors. In addition, recent work has predicted[14] and shown polarization symmetry breaking of optical pulses in fiber ring resonators with residual birefringence[15–19]. For example peak pulse powers of 2.7 W (average power of 110 mW) have been used when observing spontaneous symmetry breaking between two orthogonal polarization modes[19].

In this work, we experimentally demonstrate that Kerr-nonlinearity mediated symmetry breaking can be observed for the polarization states of continuous wave light in geometrically linear, polarization degenerate, Fabry–Pérot (FP)-type cavities at 7 mW optical power. This symmetry breaking is demonstrated for linearly polarized input light that is sent into a high Finesse fiber cavity. At low powers this system maintains symmetry such that the polarization of the cavity field matches that of the input. At a measured threshold power of 7 mW, spontaneous symmetry breaking of the resonator modes splits up the linear polarized light into left and right polarized light, with one handedness being transmitted and the other one reflected. We further demonstrate that the output polarization can be optically controlled by using a resonator with slight asymmetries due to birefringence. This enables us to continuously change the output polarization state from linear to elliptical and close to circular polarization. Together with an additional polarizer, the Kerr polarization symmetry breaking can be used to generate an orthogonal polarization component with respect to the linear polarized input light. This could find applications in all-optical polarization controllers for photonic circuits that require fast response times beyond thermally or mechanically actuated polarization controllers[20–22]. If required, the output power could be kept constant by using a subsequent amplifier that is operating in saturation. In addition, this type of polarization controller does not rely on magneto- or electro-optical effects[23,24] and only relies on the Kerr-nonlinearity that is present in all materials, thus reducing fabrication complexity and eliminating the need for electrical connections.

## Results

The polarization interactions discussed here are mathematically analogous to the Kerr interaction between counter-propagating light[11–14,25]. Thus, this effect can be similarly used for all-optical information processing and storage of information[26–30]. Integration of this system on-chip would also give enhanced sensing of polarization effects beyond shot noise limitations.

Nonlinear interactions of light are extremely weak and are normally only appreciable in high-power systems, or those in which the intensity is resonantly enhanced. The advent of high-Q ring resonators[31] and FP cavities[32] has led to extensive research in nonlinear optics at low powers and small footprints, promising application in photonic integrated circuits[33]. The temporal evolution of the electric field inside a resonator consisting of a nonlinear $\chi^{(3)}$ Kerr medium is given by Eq. (1), a modified version of the Lugiato–Lefever Equation[34], which is normalized to dimensionless quantities, ignores dispersive and fast-time effects, and is extended to include coupled polarization effects (see details in Methods section)[14,35,36]:

$$\frac{dE_\pm}{dt} = \tilde{E}_\pm - E_\pm - i\delta E_\pm + i\left(|E_\pm|^2 + 2|E_\mp|^2\right)E_\pm \quad (1)$$

in which the subscript + (–) denotes the right- (left-) handed circular polarization, the first term ($\tilde{E}_\pm$) represents the input fields that are sent into the cavity, the second term ($-E_\pm$) represents losses inside the cavity, the third term ($i\delta E_\pm$) represents the field-cavity detuning and the final term ($-i(|E_\pm|^2 + 2|E_\mp|^2)E_\pm$) represents the Kerr effect.

The final two terms of Eq. (1) are of the same form and can be taken together as an effective cavity detuning. This follows from the physical manifestation of the Kerr nonlinearity in this system as an intensity-dependent refractive index, in which the effective refractive index that a beam experiences is dependent on its own intensity via self-phase modulation (SPM), and the intensity of the cross-polarized beam via cross-phase modulation (XPM). When the Kerr effect is a result of nonresonant electronic response, as in our system, the effect of XPM is twice that of SPM, leading to the factor of 2 in Eq. (1).

The steady-state intracavity powers can be found from Eq. (1):

$$|E_\pm|^2 = \frac{|\tilde{E}_\pm|^2}{1 + \left(|E_\pm|^2 + 2|E_\mp|^2 - \delta\right)^2} \quad (2)$$

which can be understood as a tilted Lorentzian response with nonlinear effective detunings $\delta_{\text{eff},\pm} = \delta - |E_\pm|^2 - 2|E_\mp|^2$.

The symmetry of Eq. (2) can be seen by interchanging the ± indices, provided that the inputs to both modes are equal ($\tilde{E}_+ = \tilde{E}_-$ i.e. a linearly polarized input). Figure 1a shows the expected response of this system to a linearly polarized input that is frequency-scanned across a resonance without breaking of the symmetry; Kerr and thermal nonlinearities give the resonance a triangular shape[37], and the symmetry in the system ensures that both cross-polarized components of the input couple equally, preserving the polarization of the input. However, above some threshold power, and for a range of detunings, this symmetric state becomes unstable due to the differing magnitudes of SPM and XPM[14]. Under these conditions, any small difference in intensity between the two modes is amplified because the stronger mode drives the weaker mode further out of resonance via XPM. Figure 1b shows how random noise leads the system to spontaneously adopt a handedness, with one mode dominating over the other.

It appears as if this system has broken conservation of angular momentum, since the input has no angular momentum but the cavity field has a handedness. This is not the case, instead the linear input has been broken into its constituent components of opposite angular momenta, with one dominating the coupling into the cavity. The opposite handedness is reflected from, rather than transmitted through, the cavity, and thus conserves angular momentum.

This process can also be viewed as a partial conversion of the linear polarized input light into an orthogonal polarization state. The input to the cavity is vertically polarized, so has no horizontal component, however the resulting cavity/output fields are elliptical (due to the partial rejection of one circular polarization), which now must have a horizontal component that is $\pm\pi$ out of phase with the vertical component. Accordingly, the process can either be characterized by the difference in the output intensities in the circular basis, or by the generation of light in the horizontal basis.

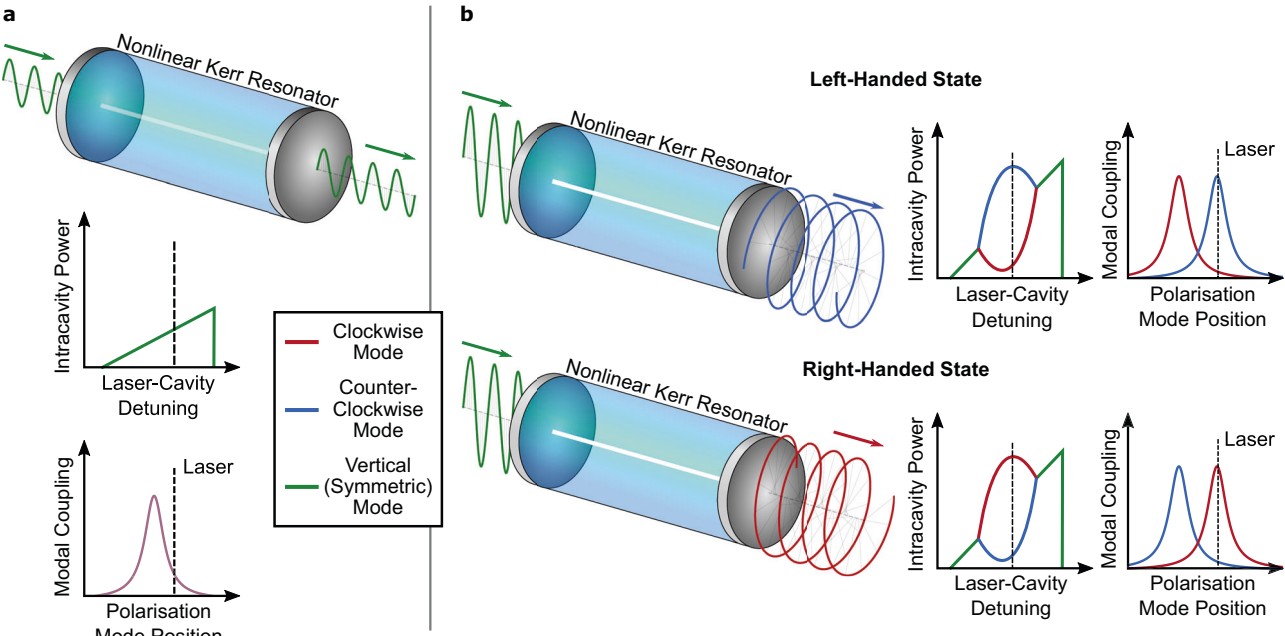

**Fig. 1 Kerr interaction of the polarization modes of light.** Linearly polarized light enters a nonlinear high-Q Fabry–Pérot cavity with degenerate polarization modes. **a** Below threshold power, the resonator equally supports all polarization states and the output polarization matches the input. **b** The linear polarized input light can be described as a superposition of left- and right-circular polarized light. Above a threshold power exists a regime in which the resonator cannot simultaneously support both left- and right-circular polarization modes. This leads to a spontaneous symmetry breaking in which the output develops an angular momentum with random handedness, even though the input light is linear polarized with zero angular momentum (momentum is conserved with the reflection of the opposite-handed light). The plots on the right in **b** show the intracavity power and resonance frequencies of the symmetry broken states.

**Experimental setup for spontaneous polarization symmetry breaking**. Figure 2 shows a schematic for the experimental setup. The cavity for this work is made from a 2-m-long single-mode silica fiber, connected at both ends to highly reflective dielectric Bragg mirrors that are coated onto fiber ends[38]. The mirrors have a reflectivity >99% at 1550 nm. Together, they form a high-finesse cavity ($F \approx 140$) with very narrow linewidths ($\delta\nu \approx 0.40$ MHz, $Q \approx 4.9 \times 10^8$). Even though the finesse is already high, these parameters could be further improved by directly depositing the mirrors on both ends of a fiber to form a cavity, minimizing the losses at the connector.

Polarization symmetry breaking requires the splitting of the resonances to be dominated by the Kerr effect; the splitting due to birefringence (a manifestation of linear coupling between polarization modes) should be minimal. In principle this resonator should show degenerate polarization modes, however asymmetries in the mirror deposition and stresses in the fiber lead to some slight amount of birefringence which cannot be neglected due to the narrow linewidth of this resonator. To observe symmetry breaking, it was found experimentally that this linear resonance splitting must be below $\approx 5\%$ of the cavity linewidth, which constrains the residual birefringence to $\delta n/n < 0.05/Q < 1 \times 10^{-10}$, where $\delta n$ is the difference in the refractive index between the two polarization modes and $n$ is their average refractive index (see Methods section for more details). The residual birefringence could be eliminated with more careful waveguide design or by using spun fibers. In this work we use an intracavity polarization controller to eliminate the residual birefringence.

In the experiment, light from a tunable diode laser is amplified by an erbium-doped fiber amplifier (EDFA), before being sent through an isolator to minimize unwanted effects from back reflections of the fiber cavity. The output polarization of the EDFA changes with power, so the power input to the cavity is instead controlled using a variable attenuator which maintains polarization across the required power range. Finally, the input polarization is set to linear by a polarization controller (PC1) before entering the cavity. This polarization state is henceforth defined to be the vertical polarization direction.

Light then enters the cavity and builds up in intensity, with some part exiting through the output port. The output signal is then split and each branch is sent to a photodiode via a polarization controller and polarization beam splitter (PBS). The polarization controllers are set to map the cavity polarization modes to the PBS basis, with one branch monitoring the opposite circular polarizations and the other monitoring the vertical and horizontal polarizations. The signals from the photodiodes can then be used for real-time monitoring of the cavity polarization state when the laser frequency is swept across a resonance.

**Experimental results and demonstration of polarization control**. The transmission through the cavity for polarization states at different input powers is shown in Fig. 3. At low powers, the linear input polarization couples equally into the left- and right-circular polarization modes of the cavity as would be expected for a system without birefringence. Correspondingly, this means that the output polarization matches the input for any cavity detuning; taking the input polarization to be vertical, there is no horizontal component to the output.

Above a threshold input power of 7 mW, spontaneous symmetry breaking is observed. This is exhibited by a difference in the output powers of opposite circular polarizations. In addition, we observe the sudden spontaneous generation of horizontally polarized light, which is the manifestation of the same phenomenon in a different basis. Larger input powers lead to a greater power splitting, consistent with the power dependence of the Kerr effect.

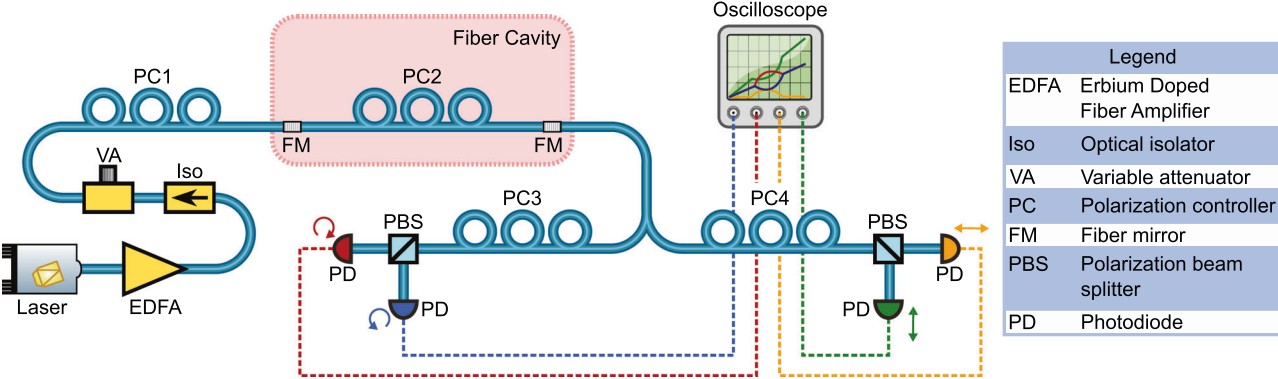

**Fig. 2 Experimental setup.** A high-finesse Fabry–Pérot fiber cavity is realized by connecting an optical fiber on both ends to fibers with dielectric Bragg mirror stacks (fiber mirror, FM). To attain degenerate polarization modes, a polarization controller (PC2) is placed within the cavity, which is used to cancel any birefringence in the fiber and mirrors. Light is sent into the cavity from a tunable diode laser via an erbium-doped fiber amplifier (EDFA) with an isolator (Iso) to prevent back reflections. A variable attenuator (VA) is then used to control the power of the input light and its polarization is set by polarization controller PC1. The output of the cavity is split by a 50:50 fiber coupler and each branch is directed to photodiodes (PD) via PC3,4 and polarization beam splitters (PBS). These final PCs are used to map the cavity's polarization states to the PBS such that the PDs each monitor a distinct polarization mode of the resonator.

**Fig. 3 Measurement of spontaneous polarization symmetry breaking. a** At low powers, both the right- (red) and left-handed (blue) polarization states couple equally into the cavity. This corresponds to the output light always having vertical polarization (green), with no horizontal (yellow) component. **b** Above threshold, spontaneous symmetry breaking changes the relative optical power in the different polarization modes in a range of cavity detunings. In this regime, there has been the spontaneous generation of horizontally polarized light, and a reduced amount of the vertical polarization. **c** The symmetry breaking increases at higher input powers. **d** Threshold behavior for the polarization symmetry breaking. The red curve shows the maximum power difference between left- and right-circular light for different input powers. The yellow curve shows the power of the generated horizontally polarized light.

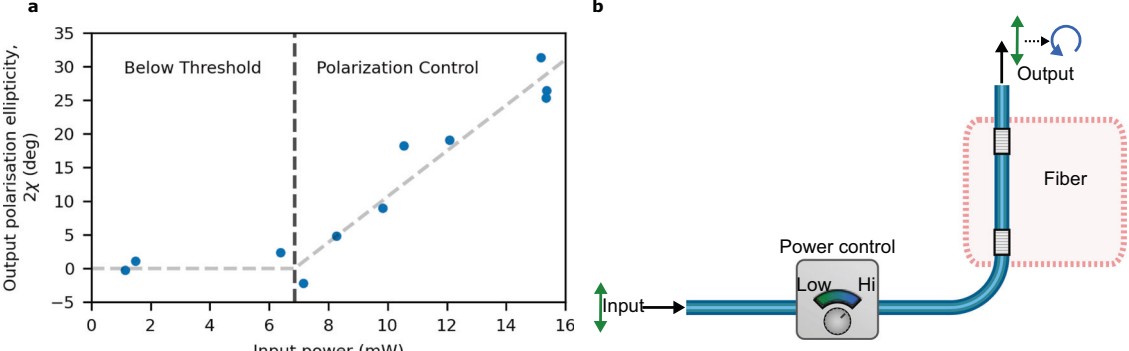

**Fig. 4 Polarization control using the Kerr effect. a** Experimental demonstration of the control of the output field's ellipticity, given by the Stokes parameter $\chi$, for different input powers. The output light remains linearly polarized ($\chi = 0$) for powers below threshold after which it becomes increasingly circular with increasing input power. **b** Concept of a Kerr polarization controller. Linearly polarized light is input into a cavity such that the output polarization can be controlled by modifying the input intensity. The cavity must be slightly biased towards one circular polarization, forcing the output to have the intended handedness rather than spontaneously developing a random handedness.

In principle, the direction of the symmetry breaking i.e. the handedness of the output light should be random for every sweep of the laser through the cavity resonance. In practice, a dominant direction was seen due to residual cavity birefringence and imperfect input polarization. A small handedness in the input light leads to a preferred symmetry breaking direction, which can be used for highly sensitive polarization sensors. In addition, residual birefringence allows one mode to couple in before the other during the laser sweep, making it dominant (this effect would then be dependent on the direction of the laser frequency sweep).

Intentionally biasing the resonator with a dominant circular polarization allows for the realization of a Kerr polarization controller. Figure 4a shows how the output field's Stokes parameter $\chi$ — a measure of the ellipticity of the polarization—can be controlled using the input power. Below a threshold power, the output light remains linearly polarized ($\chi = 0$). When increasing the input power above the threshold, the output becomes increasingly circular polarized. Eventually the output light would asymptotically reach $2\chi = 90°$, which corresponds to an increasing splitting of the different polarization modes and a bigger bubble in Fig. 3. In the measurement we observe a threshold power of around 7 mW and a maximum ellipticity of $2\chi \approx 30°$. Higher values of $2\chi$ were inaccessible due to the presence of parasitic nonlinearities—Brillouin scattering and four-wave mixing (FWM)—at higher input power. The length of the cavity here studied leads to a small free spectral range (FSR) and a high density of modes. This guarantees that there will be at least one mode that is well phase-matched for such unwanted nonlinear processes. Using shorter cavities—with large FSR—can ensure that Brillouin scattering is suppressed by the lack of resonant modes in the Brillouin gain region. Similarly, FWM can be suppressed by using dispersion engineered cavities such that neighboring modes are not equidistant in frequency (e.g. by using a cavity with normal dispersion). Even lower threshold powers for the polarization control and higher attainable values for the ellipticity could be achieved by either using waveguide materials with higher nonlinear refractive index, or by increasing the finesse of the cavity mirrors. In particular with optimized mirror fabrication techniques, we expect accessible threshold powers well below 1 mW. Our results show that a suitably biased cavity can be used to form a polarization controller (Fig. 4b) for which the output polarization is dependent on the input intensity. This system could be integrated on-chip, with the input intensity being controlled by the on-chip laser pump current or embedded semiconductor optical amplifiers. Other options for power control include the use of integrated optical attenuators, e.g. based on MEMS devices, or a Mach–Zender interferometer with a controllable phase shifter in one arm. In applications that require high polarization stability, one could envision an active control loop that feeds back to the input power in order to stabilize the output polarization state.

## Discussion

We demonstrate the spontaneous symmetry breaking of the polarization state of continuous-wave light in FP-type optical resonators. Above a threshold power of 7 mW, the Kerr effect spontaneously splits up linear polarized input light into left- and right-circular polarized light, with only one of the two polarization directions being transmitted, allowing for all-optical control of the polarization state of the output light. This effect is applicable to any high-Q Kerr resonator with sufficiently small birefringence and could be used in mm-scale fiber cavities or chip-integrated microresonators. As such, the polarization symmetry breaking offers the possibility to be used in a number of applications, most evidently as nonlinear polarization filters but also as all-optical polarization controllers and enhanced polarization sensors. A number of these devices could be cascaded to map an arbitrary input polarization to any output state based on the power and detuning of the input. Such all-optical polarization control could also be of interest for applications in environments in which electronic polarization control is not feasible or not practical. As a sensor, the bifurcation at the symmetry breaking point gives a strong sensitivity of the system to the input polarization state which could be beneficial for e.g. optical neural networks, quantum information processing, and lab-on-chip systems.

## Methods

**Cross-phase modulation between polarization modes of light**. The evolution of the electric field ($\vec{E}$) in a nonlinear resonator is given by the Lugiato–Lefever equation[34], which is here generalized to take into account polarization[35]. Neglecting dispersive effects and assuming a uniform, slowly evolving cavity field, this system can be described by the following equation:

$$\frac{\partial \vec{E}}{\partial t} = -(1 + i\delta)\vec{E} + \vec{\tilde{E}} + i\left(A\left(\vec{E} \cdot \vec{E}^\star\right)\vec{E} + \frac{B}{2}\left(\vec{E} \cdot \vec{E}\right)\vec{E}^\star\right), \quad (3)$$

where $\vec{\tilde{E}}$ is the input field and $\delta$ is the cavity detuning parameter. The nonlinear term, $A\left(\vec{E} \cdot \vec{E}^\star\right)\vec{E} + \frac{B}{2}\left(\vec{E} \cdot \vec{E}\right)\vec{E}^\star$ can be simplified by assuming that $A = B$, which is the case for nonresonant electronic responses (the source of nonlinearity

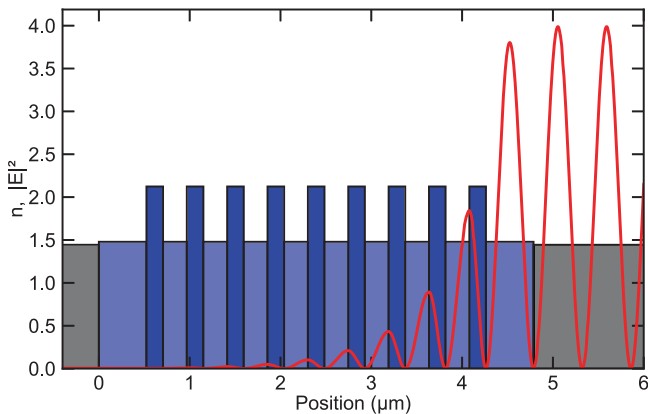

**Fig. 5 Design of the layer stack of the Bragg mirror.** The height of the bars indicates the refractive index $n$ of the silica fiber core (gray), coated $SiO_2$ (light blue) and coated $Ta_2O_5$ (blue). The red line shows the electric field intensity $|E|^2$ formed by the normalized incoming wave coming from the right side and the reflected wave.

in silica)[5,39]. The electric field $\vec{E}$ can be split into two linear components $\vec{x}$, $\vec{y}$ using

$$\vec{E} = E_x \vec{x} + E_y \vec{y}, \qquad (4)$$

which transforms Eq. (3) into

$$\frac{\partial E_{x,y}}{\partial t} = -(1+i\delta)E_{x,y} + \tilde{E}_{x,y} + iA\left(\left[|E_x|^2 + |E_y|^2\right]E_{x,y} + \frac{1}{2}\left[E_x^2 + E_y^2\right]E_{x,y}^{\star}\right). \qquad (5)$$

The nonlinear term in Eq. (5) shows equal amounts of self- and cross-phase modulation along with a phase-dependent term. The balanced effects of self- and cross-phase modulation ensure that it is not possible to break symmetry in this basis.

The circular basis is defined as

$$E_{\pm} = \frac{E_x \pm iE_y}{\sqrt{2}}, \qquad (6)$$

(which can also describe the polarization of the input fields) such that

$$\frac{\partial E_{\pm}}{\partial t} = \frac{1}{\sqrt{2}}\left(\frac{\partial E_x}{\partial t} \pm i\frac{\partial E_y}{\partial t}\right), \qquad (7)$$

which on insertion of Eq. (5) becomes

$$\begin{aligned}\frac{\partial E_{\pm}}{\partial t} &= -(1+i\delta)E_{\pm} + \tilde{E}_{\pm} + iA\left(\left[|E_x|^2 + |E_y|^2\right]E_{\pm} + \frac{1}{2}\left[E_x^2 + E_y^2\right]E_{\mp}^{\star}\right)\\ &= -(1+i\delta)E_{\pm} + \tilde{E}_{\pm} + iA\left(\left[|E_{\pm}|^2 + |E_{\mp}|^2\right]E_{\pm} + E_{\pm}E_{\mp}E_{\mp}^{\star}\right)\\ &= -(1+i\delta)E_{\pm} + \tilde{E}_{\pm} + iAE_{\pm}\left(|E_{\pm}|^2 + 2|E_{\mp}|^2\right). \end{aligned} \qquad (8)$$

In this basis, cross-phase modulation now has twice the strength of self-phase modulation and thus symmetry breaking is possible[14]. The circular polarization basis is the only one without the phase terms of Eq. (5) and thus is the natural basis to describe the optical interactions. Accordingly, the input must be linearly polarized (i.e. an equal pumping of both circular directions), and the symmetry broken state will spontaneously tend to one of the circular handed polarization states.

At steady state, Eq. (8) is zero such that

$$0 = -(1+i\delta)E_{\pm} + \tilde{E}_{\pm} + iAE_{\pm}\left(|E_{\pm}|^2 + 2|E_{\mp}|^2\right)$$
$$E_{\pm} = \frac{\tilde{E}_{\pm}}{1 + i(\delta - A(|E_{\pm}|^2 + 2|E_{\mp}|^2))} \qquad (9)$$
$$|E_{\pm}|^2 = \frac{|\tilde{E}_{\pm}|^2}{1 + \left(A(|E_{\pm}|^2 + 2|E_{\mp}|^2) - \delta\right)^2}.$$

Normalizing all intensities by $A$ yields Eq. (2).

The Stokes parameter which defines the ellipticity of light, $2\chi$, can then be calculated from

$$2\chi = \arctan\left(\frac{1}{2}\left(\frac{|E_+|}{|E_-|} - \frac{|E_-|}{|E_+|}\right)\right), \qquad (10)$$

in which $2\chi = 0$ for linearly polarized light with $|E_+| = |E_-|$, and $2\chi = \pm\frac{\pi}{2}$ for right- and left-handed circularly polarized light with $|E_-| = 0$ or $|E_+| = 0$, respectively.

**Birefringence requirement.** In order to observe symmetry breaking, it was found experimentally that the differences between the resonance frequencies for the orthogonal circularly polarized modes ($\omega_{\pm}$) must be less than $\approx 5\%$ of their line-widths ($\delta\omega_+ \approx \delta\omega_- \approx \delta\omega$).

$$|\omega_+ - \omega_-| < 0.05\,\delta\omega, \qquad (11)$$

which can be written in terms of the average resonance frequency $\omega_0$ and the cavity $Q$-factor:

$$|\omega_+ - \omega_-| < 0.05\frac{\omega_0}{Q}$$
$$\frac{|\omega_+ - \omega_-|}{\omega_0} < \frac{0.05}{Q}. \qquad (12)$$

Since the resonance frequencies are inversely proportional to their respective refractive indices (and both modes were confirmed to have the same longitudinal mode number), this becomes:

$$n_0\left|\frac{1}{n_+} - \frac{1}{n_-}\right| < \frac{0.05}{Q}$$
$$n_0\frac{|n_- - n_+|}{n_+ n_-} < \frac{0.05}{Q} \qquad (13)$$
$$\frac{\delta n}{n_0} < \frac{0.05}{Q},$$

where $\delta n = |n_+ - n_-|$ is the difference between the refractive indices for the orthogonal circularly polarized modes, and $n_0$ is their average refractive index. The last step of this derivation is valid for small values of $\delta n$, which is the case for high-$Q$ cavities. In our case, the $Q$-factor of $4.9 \times 10^8$ leads to the requirement of $\frac{\delta n}{n_0} < 10^{-10}$.

**Fiber Bragg mirror fabrication.** The Bragg mirrors are produced with an reactive ion beam sputtering (IBS) thin-film deposition process (Navigator 1100, CEC GmbH) using Xenon as a sputtering gas. The IBS technology stands out by its ability to deposit layers with exceptionally low scattering loss and low residual absorption. Tantalum pentoxide ($Ta_2O_5$, $n_H = 2.124$ at $\lambda_c \approx 1550$ nm) and silicon dioxide ($SiO_2$, $n_L = 1.479$ at $\lambda_c \approx 1550$ nm) are used as high-refractive index and low-refractive index materials, respectively. The oxides are formed by oxidation of the metallic Ta (5 N purity) and Si (9 N purity) released from the sputtering targets with a deposition rate of about 0.1 nm/s. Before deposition, the vacuum chamber is evacuated down to a level in the range of $1 \times 10^{-7}$ mbar. During the deposition the vacuum pressure doesn't exceed $2 \times 10^{-3}$ mbar and the holder of the fiber tips was heated and temperature controlled to 60 °C. No post-processing or annealing is applied to the samples after the deposition. The automated coating process is precisely controlled by broadband optical monitoring. The layerstack is build up by starting with a half-wave layer of $SiO_2$ and then 9 quarter-wave layers of $Ta_2O_5$ interleaved with 8 quarter-wave layers of $SiO_2$ and then closed with a half-wave layer of $SiO_2$. In this way the coating starts with a layer that closely matches the refractive index of the core of the fiber on which it is coated and with the one it is in contact and the half wave thickness minimizes the influence of the interface from core to coating since it is placed at a node of the standing wave formed by the incoming and reflected wave (see Fig. 5).

**Polarization controller alignment.** Single mode optical fiber supports light of arbitrary polarization, making it difficult to know the polarization state at any point along the fiber. Furthermore, stresses and bending of the fiber can change this polarization state of light during propagation. Fiber polarization controllers—a set of rotating paddles which induce birefringence—were used to set the polarization states at various parts of the experimental setup, with all fibers being taped to an optical table to prevent stress-induced birefringence changes for the duration of the data collection.

The following approach was used to adjust the polarization states during our measurements. In reference to Fig. 2, we first align the polarization controllers PC1 (input) and PC2 (intracavity). For the alignment, we sweep the laser across the resonances. This is done at sufficiently low optical power to observe the resonances as Lorentzian lines without nonlinear or thermal broadening of the modes. In general, birefringence inside the cavity leads to two polarization modes with different associated resonance frequencies. Suitable control of PC3 can show these as two separate peaks on the oscilloscope from the associated photodiodes (red and blue photodiodes in Fig. 2). By adjusting PC2 we compensate residual birefringence in the fiber cavity and move the resonance frequencies of the two polarization modes closer together, while we simultaneously ensure with PC1, that light couples equally into both polarization modes. This process is repeated multiple times until the polarization modes are close enough to be considered degenerate (i.e. the difference in the resonance frequencies is negligible and well below 5% of their linewidths). In addition, we confirm that the two polarization modes have the same longitudinal mode number by confirming the mode overlap across multiple resonance pairs within the tuning range of the laser (Toptica CTL1550—range from 1510 to 1630 nm).

In the next step, PC1 and PC3 are adjusted in order to detect the left- and right-circular polarized output light. The laser output power is set sufficiently high, such that symmetry breaking will occur for linearly polarized input light. PC3 is adjusted until the signals from the photodiodes for left- and right-circularly polarized light are as symmetric as possible. There is then an iterative process in which PC1 is

adjusted, followed by a corresponding adjustment of PC3 to keep the signals as symmetric as possible. As PC1 nears the correct state—such that its output is linearly polarized—the onset of symmetry breaking is apparent from the PD signals. Fine tuning of PC1 and PC3 is then used to optimize the system until the biggest possible symmetry breaking "bubble" is observed.

Finally, we adjust PC4 in order to detect the horizontal and vertical polarization states. For this, we define the linear input polarization state as being vertical. At input powers below the symmetry breaking threshold, the output light is expected to be vertically polarized and no horizontally polarized light exits the resonator. In the experiment we reduce the input power with a variable attenuator that does not affect the polarization state. Now, PC4 is iteratively adjusted until the signal on the horizontal photodiode vanishes. This has now mapped the cavity vertical and horizontal polarization states onto the second PBS, such that the PDs monitor these cavity components. The experiment is performed immediately after adjusting the polarization controllers to minimize polarization drifts induced by temperature changes in the laboratory.

**Reporting summary**. Further information on research design is available in the Nature Research Reporting Summary linked to this article.

## Data availability
The raw data used in this article is available from the corresponding author upon reasonable request.

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

## Acknowledgements
This work was supported by the European Union's H2020 ERC Starting Grants 756966 and 853564, the H2020 Marie Sklodowska-Curie COFUND "Multiply" 713694, the Marie Curie Innovative Training Network "Microcombs" 812818, the Max Planck Society, and the Engineering and Physical Sciences Research Council (EPSRC) via the CDTs for Applied Photonics and Quantum Systems Engineering.

## Author contributions
N.M. and P.D.H. conceived and designed the experiments. N.M., L.D.B. and S.Z. per-formed the measurements. N.M., L.D.B., S.Z., M.T.M.W., G.O., L.H. and P.D.H. analyzed the results. T.W., V.J.W., T.S., V.B. and T.H. fabricated and characterized the fiber mirrors. All co-authors contributed to preparing the manuscript.

## Funding

## Competing interests
The authors declare no competing interests.
