## [Peer review file · Nature Communications]

REVIEWER COMMENTS

Reviewer #1 (Remarks to the Author):

This manuscript, written by N. Moroney et al., demonstrates Kerr nonlinearity-induced polarization changes with an optical fiber-based high-Q Fabry-Perot resonator. By utilizing spontaneous symmetry breaking in optical resonators, which the author's group has intensively studied, an intriguing concept of an all-optical polarization controller is proposed. Polarization symmetry breaking is attracting much interest from broad audiences, including me, but the concept and applications are not well developed so far. Therefore, the idea proposed in this paper sounds quite interesting, and possibly we anticipate future application and chip-scale integration.

As mentioned above, I recognize that the concept of this paper leaves nothing to be desired. But, on the other hand, a part of experimental details remains still to be addressed before making a strong recommendation of this manuscript to be accepted.

1. The following is the most doubtful part for me by now. How did the authors get to know the polarization state for each section experimentally? In general, polarization in a single-mode optical fiber is difficult to evaluate, and furthermore, quite sensitive and changeable to the surrounding environment. A fiber-based polarization controller is useful in this respect; it changes the output polarization by stress-induced birefringence. On the other hand, accurate polarization control is not straightforward compared to free-space optics since the input polarization is not specified. I suppose that such precise polarization control is a critical factor for demonstrating this concept; however, the experimental details or procedures are hardly mentioned in the current manuscript. The authors can address this point even if they have supplement material.

2. Even though this is related to the first comment, I suggest the authors consider the use of polarization-maintaining (PM) fiber and free-space optics instead of single-mode fiber couplers and polarization controllers. For a proof-of-concept demonstration, the rest of the fiber cavity could be replaced by PM and free-space components (waveplate, etc.) to make this experiment convincing. Otherwise, it may be possible to use a polarimeter.

3. On page(4/8), " a strict requirement on the birefringence of $\sim\sim$ " What does this relation come from? It is hard to understand how the authors obtain this requirement. If it is from an experiment, the authors may clarify the details. (Δn and n should be defined here.)

4. In Fig. 3(a), the total output power seems to reach up to 10 mW (3+3+4) as the sum of right-, left- and vertical states, whereas the input power is only 5 mW. How did the authors evaluate output powers? This question applies to (b and (c as well.

5. It is just out of interest (so I'm not asking for it). Is there any supporting simulation for polarization symmetry breaking? Is it possible to reproduce the experimental results, for example, in Fig. 3? I have looked over the previous work by the authors (Ref. 11) in which theoretical results are comparable with the experimental results.

Reviewer #2 (Remarks to the Author):

I have carefully read and reviewed the manuscript by N. Moroney et al. The authors have experimentally implemented an input-power-dependent polarization controller based on a high-quality-factor Kerr nonlinear resonator. The spontaneous symmetry breaking of the polarization states

in the optical fiber resonator was demonstrated at a low threshold input power of ~ 7 mW and was also applied to an all-optical polarization controller. I expect that the demonstration and development in this paper may have the potential to be applied to a variety of fundamental studies and exciting applications, e.g., highly-sensitive polarization sensors. Thus, I believe this work could be suitable for Nature Communications, but only after addressing the following concerns and comments:

1) As the authors mentioned, spontaneous symmetry breaking is too sensitive, and thus, its direction and the handedness of the output light should be random or be determined ambiguously. Such properties would be helpful for a polarization sensor but would be disadvantages for a deterministic polarization controller. As shown in Fig. 4a, the output polarization ellipticity fluctuates considerably depending on the input power. Can the authors provide a method to stabilize the operation of the developed polarization controller or guarantee its reliable utilization?

2) In Fig. 3(d), the threshold of the spontaneous symmetry breaking seems to be ~ 5 mW. On the other hand, in Fig. 4a, the threshold of the demonstrated polarization controller is ~ 7 mW. What makes such a difference in the threshold power between two results?

3) It is also necessary to clarify the exact values of the input power employed for Figs. 3a-3c. Moreover, according to Fig. 3d, the result in Fig. 3c corresponds to the maximum power difference of < 5 mW. However, Figure 3c shows a maximum power difference of ~ 6 mW. Please present the input powers employed in Figs. 3a-3c exactly and indicate their corresponding points clearly in Fig. 3d.

4) The potential of the developed system toward highly sensitive polarization sensors is acceptable. However, the motivation and advantage of implementing an "input-power-dependent" polarization controller are ambiguous. The paper focusing on an all-optical Kerr polarization controller should have clarified the importance or relevance of developing such an input-power-dependent polarization controller.

5) Related to the comment above, the output polarization ellipticity changes from 0 to ~ 30 degrees. A promising polarization controller requires a range of changing the ellipticity approaching 90 degrees. Can the authors provide a strategy or possibility to achieve such a full-range polarization control?

6) The threshold of spontaneous symmetry breaking of the polarization state is low compared to the previously reported works in the literature. However, the introductory paragraph has not yet provided a sufficient overview and description of the related, state-of-the-art results. Particularly, it is necessary to compare the result of spontaneous symmetry breaking achieved in this paper with the results in Ref. 16 and the previous work of the authors in Ref. 22 in further detail and clarify the novelty of the present work.

I also raise the following minor comments:

1) To allow other researchers to reproduce the results and achievements in this paper, more details of the fiber resonator, fabricated mirrors, and employed equipment are required to be added (might be in the Methods section).

2) There are a few typos. For example, in Ref. 22, Phys. Rev. A. 101, 13823 (2020) should be Phys. Rev. A. 101, 013823 (2020).

Reviewer #3 (Remarks to the Author):

The work of N. Moroney et al. uses the fast Kerr nonlinearity in a high-finesse FP fiber cavity to achieve all-optical control of the polarization state. The operation principle is spontaneous symmetry breaking arising from Kerr nonlinear phase shift which amplifies an initial imbalance, whether intentionally introduced or due to drift/fluctuations.

This is a nice demonstration of the capability of Kerr resonators as an all-optical polarization controller and could be useful for certain applications. However, I do not believe Nature Communications is an appropriate journal for this work. In a previous publication of the co-authors [Woodley et al. Phys. Rev. A 98, 053863 (2018)], they already pointed out the universal nature of Kerr induced symmetry-breaking dynamics, and that the results are trivially extensible, in particular to circular polarizations of opposite handedness which is exactly what the authors show here. Furthermore, such spontaneous symmetry breaking dynamics in cavities with Kerr-like response has been known for decades [e.g. Kaplan and Meystre, Opt. Commun. 40, 229 (1982).] I do not believe this work represents important advances of significance expected in a journal of Nature Communications' caliber, but is likely suitable for a more specialized optics journal.

Additional comments:

1. Since there are multiple polarization controllers in the experimental setup and the success of the experiment is contingent on their correct settings, a detailed description on how polarization states are set and diagnosed at various points of the setup seems pertinent.
2. 6th last line before subsection 3, 'though' -> 'through'

Manuscript title: "A Kerr Polarization Controller"

We would like to thank the reviewers for their feedback, which shows a careful consideration of our manuscript, and the opportunity to respond. We have attached a revised manuscript, which we believe is now of significantly higher quality due to the valuable feedback from the reviewers. Our responses (black) to the individual reviewer comments (*blue, italics*) will follow, with reference to changes in the manuscript. We have included a **redlined** version of the manuscript with highlighted changes. We also added new parts to the "Methods Section" to describe the mirror fabrication and the procedure for aligning and measuring the polarization states.

Reviewer 1

This manuscript, written by N. Moroney et al., demonstrates Kerr nonlinearity-induced polarization changes with an optical fiber-based high-Q Fabry-Perot resonator. By utilizing spontaneous symmetry breaking in optical resonators, which the author's group has intensively studied, an intriguing concept of an all-optical polarization controller is proposed. Polarization symmetry breaking is attracting much interest from broad audiences, including me, but the concept and applications are not well developed so far. Therefore, the idea proposed in this paper sounds quite interesting, and possibly we anticipate future application and chip-scale integration.

We thank the reviewer for the constructive comments and careful review of our manuscript. A future chip-scale integration would be indeed a nice addition to the ever-growing toolbox of integrated photonic components.

As mentioned above, I recognize that the concept of this paper leaves nothing to be desired. But, on the other hand, a part of experimental details remains still to be addressed before making a strong recommendation of this manuscript to be accepted.

- 1) The following is the most doubtful part for me by now. How did the authors get to know the polarization state for each section experimentally? In general, polarization in a single-mode optical fiber is difficult to evaluate, and furthermore, quite sensitive and changeable to the surrounding environment. A fiber-based polarization controller is useful in this respect; it changes the output polarization by stress-induced birefringence. On the other hand, accurate polarization control is not straightforward compared to free-space optics since the input polarization is not specified. I suppose that such precise polarization control is a critical factor for demonstrating this concept; however, the experimental details or procedures are hardly mentioned in the current manuscript.*

The authors can address this point even if they have supplement material.

Thank you for these comments, as the reviewer has stated these are critical experimental techniques and so we have added a new part to the "Methods Section", which includes a detailed explanation of how the polarization controllers are aligned.

- 2) Even though this is related to the first comment, I suggest the authors consider the use of polarization-maintaining (PM) fiber and free-space optics instead of single-mode fiber couplers and polarization controllers. For a proof-of-concept demonstration, the rest of the*

fiber cavity could be replaced by PM and free-space components (waveplate, etc.) to make this experiment convincing. Otherwise, it may be possible to use a polarimeter.

This is an interesting idea, though it will be less flexible than our current setup. In particular, PM fibers are not perfect and will still have some cross-talk between polarization modes such that the output will not be exactly the same as the input. Using the method now detailed in our manuscript, we found that single-mode (SM) fiber along with polarization controllers gave us the ability to achieve the polarization states required for this demonstration rather than having to simply accept the performance of the PM counter-part, and the removal of mechanical influences gave suitable stability for this demonstration. This flexibility of this method also allowed us to see how the system responded to near-linearly polarized input light, with a small amount of ellipticity used to bias the response to give a deterministic polarization controller. An additional technical problem in our setup is that the mirrors were coated onto the connectors that are at the outside of the cavity with additional SM fiber leads. This additional length of SM fiber would complicate a change to a pure PM system leading to the cavity in our current setup.

3) On page(4/8), " a strict requirement on the birefringence of ~" What does this relation come from? It is hard to understand how the authors obtain this requirement. If it is from an experiment, the authors may clarify the details. (δn and n should be defined here.)

We indeed experimentally observed that the symmetry breaking can be seen when the difference in resonance frequencies induced by residual birefringence is less than ~5% of the linewidths. This is rather empirical and thus we removed "strict requirement" in the revised manuscript. We also defined δn and n in the manuscript and show the connection to the resonance frequency difference in the revised Methods section. Starting with the resonance frequency difference we get:

$$|\omega_+ - \omega_-| \ll 0.05 \times \delta\omega$$

Where ω_{\pm} are the resonance frequencies for the cavity polarization modes and $\delta\omega$ are their linewidths (assumed to be the same for both modes, as there were no appreciable differences in experimental data). The linewidth can be written in terms of the resonance frequency and Q-factor, giving:

$$|\omega_+ - \omega_-| \ll 0.05 \times \frac{\omega_0}{Q}$$

We can rewrite this as:

$$\frac{|\omega_+ - \omega_-|}{\omega_0} \ll \frac{0.05}{Q}$$

And now, as the resonance frequency is inversely proportional to the refractive index, $\omega_{\pm} \propto \frac{1}{n_{\pm}}$:

$$n_0 \left| \frac{1}{n_+} - \frac{1}{n_-} \right| \ll \frac{0.05}{Q}$$

$$n_0 \frac{|n_- - n_+|}{n_+ n_-} \ll \frac{0.05}{Q}$$

$$\frac{\delta n}{n_0} \ll \frac{0.05}{Q}$$

Where $\delta n = |n_+ - n_-|$, and n_0 is the average refractive index of the modes. This last step is valid for small relative differences in refractive index, which is the case for our system. The resonator used in the manuscript has a Q-factor of 4.9×10^8 corresponding to a requirement of

$$\frac{\delta n}{n_0} \ll 10^{-10}$$

We have clarified δn and n in the manuscript, and added the derivation above to the Methods section.

4) In Fig. 3(a), the total output power seems to reach up to 10 mW (3+3+4) as the sum of right-, left- and vertical states, whereas the input power is only 5 mW. How did the authors evaluate output powers? This question applies to (b and (c as well.

The output powers were found by calibrating the photodiode (PD) output voltages to input optical powers with the aid of a power meter. The measured values were doubled, to account for the 50:50 beam splitter, to give a measurement of the associated power leaving the cavity (rather than entering the photodiodes). We now realize that this isn't explained in the article, and doesn't correspond to the experimental schematic, leading to the apparent inconsistency. Thus, we have changed the figure accordingly and removed the factor of two from the power values.

5) It is just out of interest (so I'm not asking for it). Is there any supporting simulation for polarization symmetry breaking? Is it possible to reproduce the experimental results, for example, in Fig. 3? I have looked over the previous work by the authors (Ref. 11) in which theoretical results are comparable with the experimental results.

Theoretical simulations of the symmetry breaking agree qualitatively with the experimental data. However, the measurements are not as clean as the ones with counterpropagating light in whispering gallery microresonator shown in Ref 11. We attribute this to imperfections in the setting of the input polarizations and competing thermal nonlinearities of the cavity and the mirrors. In addition, laser power fluctuations during the sweep across the resonance could impact the measurement. Below is an analytical calculation of polarization symmetry breaking in an ideal resonator that exhibits only the Kerr nonlinearity. This graph is similar to previously published theoretical results (e.g. Phys. Rev. A 101, 013823).

Reviewer 2

I have carefully read and reviewed the manuscript by N. Moroney et al. The authors have experimentally implemented an input-power-dependent polarization controller based on a high-quality-factor Kerr nonlinear resonator. The spontaneous symmetry breaking of the polarization states in the optical fiber resonator was demonstrated at a low threshold input power of ~ 7 mW and was also applied to an all-optical polarization controller. I expect that the demonstration and development in this paper may have the potential to be applied to a variety of fundamental studies and exciting applications, e.g., highly-sensitive polarization sensors.

We thank the reviewer for the careful reading of our manuscript and the positive feedback. Below is our detailed response to the specific comments.

Thus, I believe this work could be suitable for Nature Communications, but only after addressing the following concerns and comments:

- 1) As the authors mentioned, spontaneous symmetry breaking is too sensitive, and thus, its direction and the handedness of the output light should be random or be determined ambiguously. Such properties would be helpful for a polarization sensor but would be disadvantages for a deterministic polarization controller. As shown in Fig. 4a, the output polarization ellipticity fluctuates considerably depending on the input power. Can the authors provide a method to stabilize the operation of the developed polarization controller or guarantee its reliable utilization?*

Fig 4a indeed shows some fluctuations in the polarization state of the light. Part of these fluctuations could also stem from noise in the measurement of E_+ and E_- , which is converted into the ellipticity 2χ using Eq. 10 in the manuscript,

$$2\chi = \tan^{-1} \left(\frac{1}{2} \left(\left| \frac{E_+}{E_-} \right| - \left| \frac{E_-}{E_+} \right| \right) \right)$$

For higher stability of the polarization state, one could envision to implement an active feedback to the power of the input light (e.g. by using an acousto-optic modulator or electro-optic modulator with bias control). In addition, a more compact and possibly chip-integrated setup could also eliminate noise sources in future work. We have added a sentence regarding the possibility to stabilize the polarization state with feedback onto the input power to the revised manuscript.

- 2) *In Fig. 3(d), the threshold of the spontaneous symmetry breaking seems to be ~5 mW. On the other hand, in Fig. 4a, the threshold of the demonstrated polarization controller is ~7 mW. What makes such a difference in the threshold power between two results?*

What looks like a 5-mW-threshold power in Fig 3d seems to be an artefact from the lines between the data points. The actual threshold is rather around 7 mW. Below is a figure which demonstrates this idea – the black dashed line shows some actual function which has a threshold behavior, with the red dots being samples of this function and the red line being the interpolation between them. Not sampling directly at the threshold point makes it appear – from looking solely at the red line – that the threshold is lower and less pronounced.

As described in the next point, the vertical scale of the figure has been changed and now agrees better with the threshold seen in Fig 4a.

- 3) *It is also necessary to clarify the exact values of the input power employed for Figs. 3a-3c. Moreover, according to Fig. 3d, the result in Fig. 3c corresponds to the maximum power*

difference of <5 mW. However, Figure 3c shows a maximum power difference of ~6 mW. Please present the input powers employed in Figs. 3a-3c exactly and indicate their corresponding points clearly in Fig. 3d.

Thank you for noticing this discrepancy, which is associated with Reviewer 1's point 4), and was ultimately due to us accounting for the 50:50 coupler at the output of the setup. Thus, we had multiplied the measured powers by a factor of 2 to account for the coupler. We've realized that this wasn't the best way to present the data which has been changed and is now more consistent with the setup in Figure 2.

4) The potential of the developed system toward highly sensitive polarization sensors is acceptable. However, the motivation and advantage of implementing an "input-power-dependent" polarization controller are ambiguous. The paper focusing on an all-optical Kerr polarization controller should have clarified the importance or relevance of developing such an input-power-dependent polarization controller.

Thank you for this feedback, we have added some details and references to the introduction part of the revised manuscript to clarify this. We see one potential advantage of our proposed polarization controller that it can be realized with easier accessible materials. Other polarization control methods require materials with electro- or magneto-optical effects, whereas the Kerr nonlinearity is present in all materials, such that it can work in any platform without the need for heterogeneous integration. Furthermore, other methods, as in References 19-23, can require complex fabrication techniques that are not necessary for this system, which ultimately "only" requires a waveguide-based resonator.

5) Related to the comment above, the output polarization ellipticity changes from 0 to ~30 degrees. A promising polarization controller requires a range of changing the ellipticity approaching 90 degrees. Can the authors provide a strategy or possibility to achieve such a full-range polarization control?

The ellipticity of 90° could be approached at higher resonance frequency splittings. Accordingly, more nonlinearity would lead to a greater splitting, and thus enabling a higher output ellipticity. Strategies to achieve this can be: increased input power, higher quality-factor resonators, reduced modal area, or higher nonlinear refractive index. The most straight forward way to increase the range of attainable ellipticity would be an increased finesse of the cavity mirrors. Alternatively, one could envision the integration of the system onto a chip platform using materials with higher nonlinearity. We mention this in the revised manuscript.

6) The threshold of spontaneous symmetry breaking of the polarization state is low compared to the previously reported works in the literature. However, the introductory paragraph has not yet provided a sufficient overview and description of the related, state-of-the-art results. Particularly, it is necessary to compare the result of spontaneous symmetry breaking achieved in this paper with the results in Ref. 16 and the previous work of the authors in Ref. 22 in further detail and clarify the novelty of the present work.

Thank you for this feedback. We have updated the introduction of the revised manuscript to better compare our work with the state-of-the-art.

Comments

- 1) *To allow other researchers to reproduce the results and achievements in this paper, more details of the fiber resonator, fabricated mirrors, and employed equipment are required to be added (might be in the Methods section).*

We have now included new information in the Methods section, which details the mirror fabrication and experimental procedure to align all polarization controllers.

- 2) *There are a few typos. For example, in Ref. 22, Phys. Rev. A. 101, 13823 (2020) should be Phys. Rev. A. 101, 013823 (2020).*

Thank you, this has been corrected.

Reviewer 3

The work of N. Moroney et al. uses the fast Kerr nonlinearity in a high-finesse FP fiber cavity to achieve all-optical control of the polarization state. The operation principle is spontaneous symmetry breaking arising from Kerr nonlinear phase shift which amplifies an initial imbalance, whether intentionally introduced or due to drift/fluctuations.

This is a nice demonstration of the capability of Kerr resonators as an all-optical polarization controller and could be useful for certain applications. However, I do not believe Nature Communications is an appropriate journal for this work. In a previous publication of the co-authors [Woodley et al. Phys. Rev. A 98, 053863 (2018)], they already pointed out the universal nature of Kerr induced symmetry-breaking dynamics, and that the results are trivially extensible, in particular to circular polarizations of opposite handedness which is exactly what the authors show here. Furthermore, such spontaneous symmetry breaking dynamics in cavities with Kerr-like response has been known for decades [e.g. Kaplan and Meystre, Opt. Commun. 40, 229 (1982).] I do not believe this work represents important advances of significance expected in a journal of Nature Communications' caliber, but is likely suitable for a more specialized optics journal.

We thank you for the review and your feedback on our manuscript. In terms of the mathematical description, there are indeed similarities between the polarization symmetry breaking and previous work e.g. on counterpropagating light. However, we think that there are important distinctions between them. The works that you mention are purely theoretical and do not present experimental data. The earlier work by Kaplan and Meystre focuses on the interaction of counterpropagating light and does not mention polarization as a potential avenue for observing these dynamics. In our theory paper (Woodley et al, Phys. Rev. A 98, 053863, 2018), we briefly mention the prospects of using the Kerr symmetry breaking for polarization control in the outlook, but without any experimental data to support this idea. Here we present the experimental demonstration for this concept. In particular the requirement for a cavity with sufficiently high finesse makes this demonstration non-trivial.

The value of this work, we believe, rather than simply being a demonstration of symmetry breaking in a new platform, is the presentation of a device for polarization control (and potentially sensing) that could find applications in many research areas and thus be interesting for a wide audience. We

understand that there is no new math required to describe the operation of this device, which explains our focus on experimental details in this manuscript.

In particular, we believe that a simple system based on a Fabry-Pérot Kerr-cavity that facilitates sensing and control of polarization states could for example be very valuable for lab-on-a-chip devices in the biophysics community. Other potential audiences include the photonics community that could use Kerr polarization control in photonic circuits for telecom applications. Especially the recent advances in standardized fabrication processes for photonic integrated circuits could make this Kerr-effect based polarization control widely accessible.

Additional comments:

- 1) Since there are multiple polarization controllers in the experimental setup and the success of the experiment is contingent on their correct settings, a detailed description on how polarization states are set and diagnosed at various points of the setup seems pertinent.*

Thank you for this feedback, we have included new parts in the Methods section which describes in detail how we are controlling and measuring the polarization states. The correct procedure to set and measure the polarization states is indeed very important for the measurements.

- 2) 6th last line before subsection 3, 'though' -> 'through'*

Thank you, this has been corrected.

REVIEWER COMMENTS

Reviewer #1 (Remarks to the Author):

I want to express my appreciation for reflecting comments on the manuscript. I believe that a newly added Method section improves the quality of the manuscript.

The following are my subsequent concerns regarding the experimental method.

I understood that the authors first performed the alignment of the polarization controllers PC1 (input) and PC2 (intracavity) to let the two distinct peaks degenerate. However, if the birefringence induced splitting is less than 5% of the linewidth, I cannot believe they look like the resonance split in transmission. It sounds to me like this method is technically hard to distinguish such close two different resonances.

The attached image shows a simple calculation to confirm my concern. The blue line is the original resonance, and the red one is shifted 5% of its linewidth from the original one. The yellow, purple, and green lines correspond to 10, 50, and 100%-shift, respectively. So I consider only 5% change is not likely to be seen as a slight broadening rather than splitting.

Reviewer #2 (Remarks to the Author):

I have carefully read the responses from the authors and the revised manuscript. The authors fully addressed the comments and concerns that I had raised in a point-by-point manner. I am satisfied with the responses and revised manuscript and believe that the results in this paper will be an interesting and important piece in the related research fields. I thus recommend the publication of the paper in Nature Communications.

Reviewer #3 (Remarks to the Author):

Initially, I was not in favor of this manuscript being published in Nature Communications. But the authors have convinced me that an experimental demonstration of nonlinear polarization control in the setting of a high-finesse resonator is not trivial and deserves to be acknowledged. Furthermore in the revision, the authors have put in much more details about their experiment, particularly those concerning the correct setting of polarization states at various points of their experimental setup. These details would undoubtedly facilitate the reproduction of the results.

Before I recommend the publication of this manuscript, however, please consider addressing the following points

1. This point is related to a comment raised by Reviewer 2. In Fig. 4a, is there a reason why the authors do not consider input powers higher than 16 mW? Assuming that the linear trend persists, it should not take an outrageous power level to reach the fully circularly polarized state and it should be well within the capability of an EDFA.
2. As a matter of fact, does the linear trend persist? In other words, given the nonlinear nature of the effect, does the width of the 'bubble' increase linearly with the pump power? Please comment.
3. The output polarization state depends on the input power for a given setup. How do the authors suggest to control the input power in an integrated system?
4. Although both Eq. 1 and 8 are correct, they are the complex conjugate of each other. Please unify.
5. In the last paragraph before discussion, "becomes increasingly circularly polarized ($\chi \rightarrow 90$)" should it not be $2\chi \rightarrow 90$?

Response to Reviewers, Manuscript ID: **NCOMMS-21-14941**

Manuscript title: "A Kerr Polarization Controller"

We would like to thank the reviewers for their feedback, which shows a careful consideration of our manuscript, and the opportunity to respond. We have attached a revised manuscript, which is **redlined** to highlight the changes. Below are our responses (black) to the individual reviewer comments (blue, italics).

Reviewer 1

I want to express my appreciation for reflecting comments on the manuscript. I believe that a newly added Method section improves the quality of the manuscript.

We thank you for the kind words and feedback that led to these improvements.

The following are my subsequent concerns regarding the experimental method.

I understood that the authors first performed the alignment of the polarization controllers PC1 (input) and PC2 (intracavity) to let the two distinct peaks degenerate. However, if the birefringence induced splitting is less than 5% of the linewidth, I cannot believe they look like the resonance split in transmission. It sounds to me like this method is technically hard to distinguish such close two different resonances.

The attached image shows a simple calculation to confirm my concern. The blue line is the original resonance, and the red one is shifted 5% of its linewidth from the original one. The yellow, purple, and green lines correspond to 10, 50, and 100%-shift, respectively. So I consider only 5% change is not likely to be seen as a slight broadening rather than splitting.

Thank you for this comment, our description in the manuscript might have been a bit unclear about the exact method that allowed us to see small birefringence induced mode splittings. Rather than monitoring the total cavity output field using a single photodetector, we separately monitored two polarisation components – the two that come after PC3 in Fig. 2.

Suitable control of PC3 then gives us two separate Lorentzian traces on the oscilloscope, one for each polarisation mode of the cavity. These two Lorentzian traces look very much like the image you have kindly attached. In these measurements, a splitting of ~5% is still quite easily visible and can thus be further reduced. In most measurements we compensated the static birefringence induced splitting to far less than 5% of the linewidths, as can be seen in Fig. 3a, upper panel.

We have clarified this also in the methods section of our revised manuscript.

Reviewer 2

I have carefully read the responses from the authors and the revised manuscript. The authors fully addressed the comments and concerns that I had raised in a point-by-point manner. I am satisfied with the responses and revised manuscript and believe that the results in this paper will be an interesting and important piece in the related research fields. I thus recommend the publication of the paper in Nature Communications.

Thank you very much for the review and the positive feedback.

Reviewer 3

Initially, I was not in favor of this manuscript being published in Nature Communications. But the authors have convinced me that an experimental demonstration of nonlinear polarization control in the setting of a high-finesse resonator is not trivial and deserves to be acknowledged. Furthermore in the revision, the authors have put in much more details about their experiment, particularly those concerning the correct setting of polarization states at various points of their experimental setup. These details would undoubtedly facilitate the reproduction of the results.

We thank you for your careful consideration of our revised work.

Before I recommend the publication of this manuscript, however, please consider addressing the following points

1. This point is related to a comment raised by Reviewer 2. In Fig. 4a, is there a reason why the authors do not consider input powers higher than 16 mW? Assuming that the linear trend persists, it should not take an outrageous power level to reach the fully circularly polarized state and it should be well within the capability of an EDFA.

It is indeed possible to reach higher powers with our EDFA. However, at higher power levels above 16 mW we observed additional nonlinearities, for example stimulated Brillouin scattering and four-wave mixing. This is mostly related to the relatively long cavity that was required for the intracavity polarization controller. The long cavity leads to a small FSR with a very dense mode spectrum, which guarantees that there will be at least one mode that is well phase-matched for parasitic nonlinearities. This could be avoided in the future with shorter (possibly chip-based) cavities with larger FSRs, such that Brillouin modes are quenched by not being resonant. In a similar way, four-wave mixing can be avoided by using a cavity with sufficiently high normal dispersion, such that the neighbouring cavity modes are not equidistant with respect to the input laser and thus prevent four-wave mixing from occurring. We have added this information also to the article.

2. As a matter of fact, does the linear trend persist? In other words, given the nonlinear nature of the effect, does the width of the 'bubble' increase linearly with the pump power? Please comment.

On a larger input power range, the system responds nonlinear such that 2χ approaches 90° asymptotically. However, for the range of powers that are investigated in our work, it follows a close to linear response. The bubble width (defined as the maximum difference in the powers of each component for a given input power) is shown as a function of input power in the following plot and it indeed follows a linear trend after the initial nonlinear section right after the onset of symmetry breaking. The initial abrupt opening of the bubble happens within a comparably small intracavity power range, such that our measurements show a direct transition from 0 to an approximately linear slope.

3. The output polarization state depends on the input power for a given setup. How do the authors suggest to control the input power in an integrated system?

In an integrated system, we could imagine a few different approaches to control the power. One way would be to control the electric current of on-chip lasers. Alternatively, one could control the current of embedded semiconductor optical amplifiers (SOAs). Other options for power control include the use of integrated optical attenuators, eg. based on MEMS devices, or a Mach-Zehnder interferometer with a controllable phase shifter in one arm. We added this information as an outlook to the discussion.

4. Although both Eq. 1 and 8 are correct, they are the complex conjugate of each other. Please unify.

Thanks for this comment. We have changed this in the revised manuscript.

5. In the last paragraph before discussion, "becomes increasingly circularly polarized ($\chi \rightarrow 90$)" should it not be $2\chi \rightarrow 90$?

Thanks, this should indeed be 2χ . We have corrected this in the manuscript.

REVIEWERS' COMMENTS

Reviewer #1 (Remarks to the Author):

I am satisfied with the author's responses and believe that the manuscript will appear in Nature Communications soon.

Reviewer #3 (Remarks to the Author):

I am happy to recommend the publication of the revised manuscript.